# Molecular Interaction Mechanism and Preservative Effect of Lactone Sophorolipid and Lactoferrin/*β*-Lactoglobulin Systems

**DOI:** 10.3390/foods12081561

**Published:** 2023-04-07

**Authors:** Yanrong Chen, Mingyuan Li, Jing Kong, Jie Liu, Qian Zhang

**Affiliations:** School of Chemistry and Chemical Engineering, Liaocheng University, Liaocheng 252059, China

**Keywords:** lactone sophorolipid (LSL), lactoferrin (LF), *β*-lactoglobulin (*β*-LG), interaction mechanism, preservative effect

## Abstract

Multispectral and molecular docking methods were used to study the interaction mode and mechanism of two important components of whey proteins, lactoferrin (LF) and *β*-lactoglobulin (*β*-LG), and of a lactone sophorolipid (LSL) mixed system. The preservation effect of the mixed system on milk was also studied and compared. The results showed that the quenching mechanism of LSL on both *β*-LG and LF was static, but that the non-covalent complexes formed were the result of the different interacting forces: hydrogen bonds and the van der Waals force for the LSL-*β*-LG system, and electrostatic force for the LSL-LF system. The binding constants of LSL-*β*-LG and LSL-LF were all relatively small, and the interaction of LSL with *β*-LG was stronger than its interaction with LF. After adding *β*-LG, LF, or the mixed system with LSL to the milk, the stability of milk emulsion was effectively improved in all cases, while the preservative ability was effectively enhanced only by the addition of LF or LSL-LF. These results provide supportive data and a theoretical basis for enhancing the production of dairy products and other byproducts.

## 1. Introduction

Whey protein is nutritious, easy to absorb, and contains a variety of active ingredients. Beta-lactoglobulin (*β*-LG) is the main whey protein in milk, accounting for about 50% of the whey protein. It performs a variety of biological activities and functions, and is able to easily interact with small hydrophobic ligands [1,2]. The polypeptide chain generated by the hydrolysis of *β*-LG has antihypertensive, antiviral, antibacterial, and hypoglycemic effects [3,4,5]. Lactoferrin (LF) is an important non-heme iron-binding glycoprotein in whey protein, accounting for about 26% of breast milk whey protein, and was first reported nearly 50 years ago [6,7,8]. LF is a highly multifunctional protein that participates in a variety of physiological functions, and performs various biological activities such as iron metabolism and antibacterial, antiviral, immunomodulatory, and antioxidant functions [9,10].

Given these significant functions, *β*-LG and LF are usually selected as natural additive agents, and considerable research has been conducted on their interaction with natural small molecules. The investigation of the interaction of three polyphenols (chlorogenic acid, ferulic acid, epigallocatechin-3-gallate) with *β*-LG has provided some guidance for the processing, production, and consumption effects of functional dairy products [11]. The non-covalent interaction between *β*-LG and luteolin by ultrasound has been shown to regulate the human intestinal microbiome and conformational epitopes, which reduces the risk of allergy [12]. The studied interaction of *β*-LG with rutin at physiological pH has important implications for the food industry, clinical medicine, and life sciences [13]. The binding of *β*-LG to curcumin derivatives could help design drugs and other bioactive molecules [14]. The complex coacervates of bovine LF and sodium alginate were found to protect the structure and related functions of LF during gastric digestion [15]. Therefore, given the unique structure of LF and oat *β*-glucan nanocomposites, they are considered suitable for wide use in the food, pharmaceutical, and cosmetic industries [16].

As a special kind of amphiphilic molecule co-composed of hydrophilic and hydrophobic parts, surfactants can be found in many applications from laboratory to commercial products, such as in the fields of pharmaceuticals, cosmetics, food science, paint, nanotechnology, petroleum recovery, bioremediation, chemical transformation, and drug delivery. In recent years, the application of surfactant formulations in the global market has increased steadily [17]. Moreover, during recent global epidemics, relevant personnel have reported the application of surfactants in vaccine formulation and lung failure prevention [18]. It has been demonstrated that surfactants can help the formation of helices in some proteins, thereby promoting protein structure formation. There is also extensive research on the use of surfactants to solubilize drugs and pharmaceuticals. Therefore, it is evident that the interaction between surfactants and macromolecules is important for many applications, including environmental processes and the pharmaceutical industry [19]. It was reported that the denaturation of the model protein apo-*α*-lactalbumin was induced by acidic and lactonic SL through weak and saturable protein–surfactant interactions [20]. Hansted et al. used spectroscopic and calorimetric techniques to elucidate how surfactants interacted with bovine *β*-LG and modulated its heat-induced aggregation, which suggested that surfactants may be a practical way to modulate whey protein properties [21]. By tracking the dynamic steps involved in *β*-LG unfolding and refolding in the presence of SDS and C_12_E_8_, the surfactant-mediated unfolding and refolding of proteins proved to be complex processes [22]. Among numerous surfactants, biosurfactants as a kind of mild surfactant have a giant application potential in the food and medicine industries [23], and the study of their interaction with proteins has a high application value [24,25,26,27]. Sophorolipids (SLs), as a type of glycolipid biosurfactant, have also been widely investigated. Kurtzman et al. summarized the biosynthetic pathway and classification of SLs [28]. Pal et al. provided a detailed summary regarding SL’ advantages (biodegradability, biocompatibility, digestibility, surface and interface activity, raw material availability, acceptable production economics, specificity, and efficiency) and disadvantages (detrimental effects to physiological systems, expensive large-scale production, difficulty obtaining pure substances, very low productivity, strong foam formation leading to decreased production yield). They summarized the application of SLs in antimicrobial and/or anti-biofilm effects against different bacterial strains including *Starmerella bonmicola*, *Candida albicans*, *Candida* (*torulopsis*) *apicola*, *Candida batistae*, etc. [29]. In our previous studies, the addition of LSL has been shown to change the secondary structure of soy protein isolate (SPI) and collagen oligopeptides (COP), and to improve some of their physical and chemical properties (stability, foaming, and emulsifying properties) [27,30]. However, there is little research on mixed systems of biosurfactants and animal or plant proteins such as *β*-LG and LF. In order to observe and compare the performance of these two proteins with added biosurfactants [31,32], the interactions of *β*-LG and LF with LSL under the same conditions were respectively investigated and compared in the current study, and their interaction forces and mechanisms were explored using fluorescence, UV-vis, CD, and MD simulations. The preservation effects of the two systems on milk were compared under the same conditions. These findings will help us to compare the binding behaviors of LSL on *β*-LG and LF, and their application in dairy products. The results will provide a confirmed experimental and theoretical basis for the food industry.

## 2. Materials and Methods

### 2.1. Materials

*β*-LG with a purity of ≥0.900 was purchased from Beijing Jinming Biotechnology Co., Ltd. (Beijing, China), and LF with a purity of 0.950 was purchased from YuanYe Biotechnology Co., Ltd. (Quanzhou, China). LSL with a purity of ≥0.900 was purchased from Xi’an ZhengLi Biotechnology Co., Ltd. (Xi’an, China), and the structure is shown in Appendix A. In addition, 2,3,5-Triphenyltetrazolium chloride (0.5% sterile TTC solution) and lecithin-Tween 80 nutrient agar were purchased from Guangdong HuanKai Biotechnology Co., Ltd. (Guangzhou, China); 0.9% sodium chloride solution (physiological saline) was purchased from ChenXin Pharmaceutical Co., Ltd. (Jining, China); and milk was provided by JiaBao Dairy Co., Ltd. (Jinan, China). All other reagents and materials used were of analytical grade and were supplied by Tianjin Chemical Reagent Co., Ltd. (Tianjing, China).

### 2.2. Solution Preparation

An appropriate amount of *β*-LG, LF, or LSL powder was weighed and dissolved in 10 mmol/L phosphate buffered solution (PBS) with a pH of 7.4. The prepared solutions were stirred at 25 C for 2 h, and then stored at 4 C overnight to achieve complete dissolution. The *β*-LG/LSL or LF/LSL mixed system was prepared by mixing the above *β*-LG, LF, and LSL solutions, and diluted using PBS to the different concentrations.

### 2.3. Fluorescence Spectroscopy

The spectra were measured using an F-7100 fluorescence spectrometer (Hitachi, Japan) and the temperatures were controlled at 288.2 K, 298.2 K, and 308.2 K using a water circulator (±0.1) K. The excitation wavelength was 280 nm, and the excitation and emission slit widths were both fixed at 5 nm. The measurement parameters of the synchronous fluorescence spectra: the scan wavelength was in the range of 260~340 nm when Δ*λ* = 15 nm and the scan wavelength was in the range of 220~360 nm when Δ*λ* = 60 nm. The three-dimensional (3D) fluorescence spectra had a scanning range of excitation and emission wavelength set at 200~500 nm.

### 2.4. Ultraviolet-Visible (UV-vis) Adsorption Spectroscopy

The measurements were performed at 298.2 K using a UH4150 spectrophotometer (Hitachi, Japan). The spectra were blank corrected using LSL solution and the wavelength ranged from 250 nm to 320 nm.

### 2.5. Circular Dichroism (CD) Spectroscopy

The measurements were performed on a Jasco J-810 CD spectrometer (JASCO, Japan) at 298.2 K. The scan wavelength was in the range of 190~250 nm and the scan rate was 100 nm/min.

### 2.6. Fourier Transform Infrared (FT-IR) Spectroscopy

A FT-IR spectrometer (Nicolet 5700, Thermo Scientific, Waltham, MA, USA) was used to measure the spectra at 298.2 K. The wave numbers ranged from 500 cm^−1^ to 4000 cm^−1^.

### 2.7. Dynamic Light Scattering (DLS) Measurement

The *β*-LG and LF with or without LSL were prepared using milk emulsion or PBS as solvent, then the mean hydrodynamic diameter (*D*_h_) was measured using a Zetasizer Nano ZS (Malvern Panalytical Ltd., Malvern, UK) at 298.2 K. Each sample was tested at least 3 times, and the test results were directly obtained through the Zetasizer Nano software.

### 2.8. TTC Colony Color Test

The milk was diluted with normal 10 times saline solution, then the diluted milk (20 μL) was injected into 1 mL sterilized dishes. For the convenience of observing colonies, 1 mL 0.5% TTC solution was added to 200 mL lecithin Tween-80 nutrient agar. Next, 8 mL nutrient agar solution was poured into the dish, and was melted and cooled to 318~323 K. Then we immediately rotated the plate to thoroughly mix the sample and medium. After the agar was solidified, the plate was turned over and cultured in a 310 K incubator for 48 h. In the same environment, colonies were first observed with the naked eye to count the number of colonies, and then checked with a magnifying glass magnified 10 times to prevent omission.

### 2.9. Molecular Docking

The docking approach was used to study binding interactions of *β*-LG-LSL and LF-LSL, which were simulated using AutoDock 4.2.6 software. The 3D structures of *β*-LG (PDB: 3NPO) and LF (PDB: 1BIF) were downloaded from the PDB database (http://www.rcsb.org/, accessed on 10 November 2022). The 3D structure of LSL was obtained from the online database PubChem (https://pubchem.ncbi.nlm.nih.gov/, accessed on 10 November 2022). The interaction between *β*-LG-LSL and LF-LSL was visualized by VMD software.

## 3. Results and Discussion

### 3.1. Interacting Mechanism of LSL on β-LG or LF by Spectroscopic Method

#### 3.1.1. Fluorescence Quenching Spectroscopy

At the excitation wavelength of 280 nm, two tryptophan residues (Trp19 and Trp61) and four tyrosine residues (Tyr20, Tyr40, Tyr99, and Tyr102) of *β*-LG or LF were excited [33], appearing to reach an obvious peak near 340 nm. The fluorescence intensity of *β*-LG decreased continuously with the increase of LSL concentration, as shown in Figure 1a, and the maximum emission wavelength did not shift significantly. A similar trend was found at the temperatures of 288.2 K and 308.2 K, as shown in Appendix A. This indicates that the formation of *β*-LG-LSL complexes have little influence on the microenvironment around the amino acid residues, but induce a decrease in fluorescence intensity. For the protein of LF, its fluorescence intensity also decreased with increasing LSL concentration (Figure 1b), and a certain degree of blue-shift occurred at the temperatures of 288.2 K, 298.2 K, and 308.2 K, which had blue-shifted by 6 nm, 4 nm, and 2 nm respectively. This indicates that the microenvironment around the amino acid residues becomes more hydrophobic and the polarity becomes smaller due to the interaction of LF and LSL.

Ordinary fluorescence spectroscopy cannot distinguish between the overlapping fluorescence peaks of Trp and Tyr in proteins, while synchronous fluorescence spectroscopy has the advantages of high sensitivity and good selectivity. At Δλ = 15 nm and Δλ = 60 nm, characteristic information of Tyr and Trp residues is provided, respectively [34]. For both systems, the fluorescence intensities of the Tyr and Trp residues were significantly quenched with increasing LSL concentration, as shown in Figure 2c–f. It can be seen that the fluorescence intensity quenching of Trp (Δλ = 60 nm) was stronger than that of Tyr (Δλ = 15 nm), indicating that Trp contributes more to the intrinsic fluorescence quenching of *β*-LG or LF than Tyr. In other words, the binding site is closer to the Trp residues. Furthermore, with the LSL addition, the maximum emission wavelength of Trp residues in *β*-LG or LF did not change significantly, while the wavelengths of Tyr residues in *β*-LG and LF were blue-shifted by 10 nm (from 297 nm to 287 nm) and 4 nm (from 292 nm to 288 nm), respectively. In detail, a clear blue shift was induced by 0.1 g/L LSL addition, but the wavelength underwent little change during increased LSL concentrations. In other words, LSL had a significant effect on the microenvironment of protein, but the magnitude of the LSL concentration did not further increase this effect. In summary, the presence of LSL changes the microenvironment around Tyr, weakens the polarity of *β*-LG or LF, and results in a change in protein conformation.

For fluorescence quenching, dynamic and static quenching are two common quenching mechanisms. Dynamic quenching often occurs due to molecular collisions, while static quenching often occurs due to complex formation [35]. In order to determine the quenching mechanism of *β*-LG or LF and LSL, the fluorescence quenching data of the system were calculated by the Stern–Volmer equation [36]:(1)$$\frac{F_{0}}{F}=1+K_{q}\tau_{0}[Q]=1+K_{\mathrm{SV}}[Q]$$
where *F* and *F*_0_ represent the fluorescence intensity of *β*-LG/LF with and without LSL; [*Q*] is the LSL concentration; and *τ*_0_ is the mean fluorescence lifetime without LSL (usually 10^−8^ s). Consequently, the Stern–Volmer quenching constant (*K*_sv_) and the rate constant (*K*_q_) of the bimolecular quenching process can be obtained from the linear regression curve of *F*_0_/*F* versus [*Q*] in Appendix A. The obtained constants of *K*_q_ and *K*_sv_ at different temperatures are listed in Table 1. The data show that the *K*_sv_ values of both *β*-LG and LF decrease with the increase of temperature, and the values of *K*_q_ exceed the maximum quenching constant reported (2 × 10^10^ L·mol^−1^) [37], indicating the static quenching mode of the LSL and *β*-LG or LF combination.

During static quenching, the number of binding sites (*n*) and the binding constant (*K*_a_) are calculated according to the double logarithmic Stern–Volmer equation [38]:(2)$$\mathrm{lg}\frac{F_{0}-F}{F}=\mathrm{lg}K_{a}+n\mathrm{lg}[Q]$$

From the linear regression curve of $\mathrm{lg}\left(F_{0}-F\right)/F$ versus lg[*Q*] in Appendix A, the parameters of *K*_a_ and *n* are obtained and listed in Table 1. The *K*_a_ value decreased with the increase of temperature, meaning that the binding affinity decreased with the increase of temperature, and the binding process of the two proteins and LSL was exothermic. At the same time, this relationship between *K*_a_ and temperature once again verifies that the quenching mechanism of LSL on the two proteins is static quenching [39]. Through the comparison of the binding constants of the two proteins and LSL, it was found that the binding strength of LSL to *β*-LG was stronger than LSL to LF. However, the binding constant for the LF-LSL system had a smaller changing rate with temperature than that of the *β*-LG-LSL system, which demonstrated that the formed *β*-LG/LSL complexes were less affected by temperature and more stable than the LF/LSL complexes.

The values of enthalpy change (Δ*H*_m_), entropy change (Δ*S*_m_), and free energy (Δ*G*_m_) of the system were calculated by the Van’t Hoff Equation (3) and Gibbs–Helmholtz Equation (4):(3)$$\ln K_{a}=-\frac{\Delta H_{m}}{RT}+\frac{\Delta S_{m}}{R}$$
(4)$$\Delta G_{m}=-RT\ln K_{a}=\Delta H_{m}-T\Delta S_{m}$$

From the slope and intercept of ln*K*_a_~1/*T* (Appendix A), Δ*H*_m_ and Δ*S*_m_ were obtained, and then Δ*G*_m_ was obtained. The negative Δ*G*_m_ and Δ*H*_m_ values of both systems indicate that the binding LSL molecules on two proteins are spontaneous and exothermic. Generally, the interactions between ligands and proteins can be described by four weak forces: hydrogen bonding, van der Waals forces, and electrostatic and hydrophobic interactions. The binding forces of *β*-LG to LSL are mainly van der Waals forces and/or hydrogen bonds due to the negative values of both Δ*H*_m_ and Δ*S*_m_; and the binding of LF to LSL is mainly electrostatic attraction due to the negative value of Δ*H*_m_ and the positive value of Δ*S*_m_ [40]. Moreover, the Δ*G*_m_ value for *β*-LG-LSL system is more negative than that of LF at the same temperature, so the LSL molecules show thermodynamics superiority on binding to *β*-LG.

The conformational changes of *β*-LG and LF induced by LSL were also studied by 3D fluorescence spectroscopy. Through the 3D spectra in Figure 2, specific data are obtained and shown in Table 2. Peak *a* is the fluorescence spectrum peak of Trp and Tyr residues, and peak *b* is related to the secondary structure of protein reflecting the structural characteristics of polypeptide chain skeleton. With the LSL addition, the fluorescence intensity of peak *a* and peak *b* (*F*_0_ in Table 2) of *β*-LG and LF had a significant decrease, suggesting that LSL not only affected the microenvironment around Trp and Tyr residues to result in the change of protein conformation, but also affected the relative content of the secondary structure of proteins. In addition, after adding LSL, the *F*_0_ value at peak *a* decreased by about 72.2% and 80.0% for *β*-LG and LF; and the *F*_0_ value at peak *b* decreased by about 95.0% and 89.3% for *β*-LG and LF. These data mean that LSL had a greater effect on the microenvironment around Trp and Tyr residues of LF than *β*-LG, and LSL had a greater effect on the secondary structure of *β*-LG than LF.

#### 3.1.2. UV-vis Adsorption Spectroscopy

Originating from the absorption of UV light by tyrosine, tyrosine, phenylalanine (Trp, Tyr, Phe) residues, and peptide bonds, the UV absorption of one protein is easily changed by protein conformation due to the interaction with small molecules. As shown in Figure 3a,b, before the addition of LSL, *β*-LG and LF had distinct peaks around 280 nm, which were the absorption peaks of the conjugated double bonds of Tyr and Trp residues. With the addition and increase of LSL concentration, the absorbance of both proteins increased gradually, indicating that the formation of the LSL/*β*-LG and LSL/LF complexes led to the exposure of Trp residues and Tyr residues in the protein molecules. This exposure changed the protein conformation and was favorable for the π-π* transition of aromatic heterocycles in Trp and Tyr residues, thus enhancing the UV adsorption intensity.

In addition, the maximum emission wavelengths of both *β*-LG and LF were blue-shifted from 279 nm to 267 nm, which indicated that the microenvironment of aromatic amino acid residues of both proteins changed, with increased hydrophobicity and decreased polarity. This method can also be used to determine the quenching type of small molecules and proteins: small molecules with dynamic quenching mode generally do not affect the UV absorption spectra of proteins, while with static quenching mode they can cause changes in the absorption spectra [41]. Therefore, the quenching mechanism of both *β*-LG and LF by LSL are static quenching, which is consistent with the result of fluorescence.

#### 3.1.3. CD Spectroscopy

The effects of LSL on *β*-LG and LF secondary structures were commonly confirmed by the far-UV CD spectrum. We detected the CD spectra of *β*-LG and LF in the absence or presence of LSL, respectively, and quantitatively analyzed the secondary structure of the two proteins using SELCON3 online software. As can be seen from Figure 3c,d, both *β*-LG and LF showed negative peaks located at 216 nm, an indication of the proteins with rich *β*-sheet content [42,43], which was consistent with the results calculated by the software. The *β*-sheet content was 38.6% and 34.6% for *β*-LG and LF. After the addition of LSL, the shape and position of the peak did not change significantly, indicating that the secondary structure of *β*-LG and LF was still dominated by *β*-sheet structure. The *β*-sheet content changed to 41.0% and 33.9% for *β*-LG and LF, accompanied by the downward movement of the spectral line.

Through the combination of LSL and *β*-LG, the *α*-helix structure decreased by 18.5% (from 20.5% to 16.7%) and the *β*-sheet structure increased by 5.9% (from 38.6% to 41.0%). It is therefore speculated that the binding of LSL induces part of the *α*-helix structure to the *β*-sheet structure, which is same with the systems of three polyphenols [chlorogenic acid (CGA), ferulic acid (FA), and epigallocatechin-3-gallate (EGCG)] and *β*-LG [11]. Since the *β*-sheet structure is stabilized primarily by hydrogen bonding [44], LSL enhances the hydrogen bonding in the *β*-LG molecule. Regarding the data changes of LF and LF-LSL systems, the secondary structures showed no significant change: only a small increase of *α*-helix (from 17.2% to 18.8%) and random coil structure (from 26.9% to 27.5%), and a small decrease of *β*-sheet (from 34.6% to 33.9%) and *β*-turn structure (from 21.3% to 19.8%) were noted. It seems that the binding of LSL on LF has a better maintenance effect on the *α*-helix structure. These results confirm the binding interaction between LSL and the two proteins, but with the effect of LSL on the secondary structure of *β*-LG greater than its effect on LF.

#### 3.1.4. FT-IR Spectroscopy

The FT-IR spectra of *β*-LG and LF influenced by LSL addition are shown in Figure 3e. For *β*-LG and LF, the characteristic absorption peak of amide I band was located at 1645 cm^−1^ and 1643 cm^−1^, respectively, which was attributed to the stretch vibration of C=O, indicating that the secondary structure of *β*-LG and LF was dominated by *β*-sheet structure [45]. Comparing the spectra of proteins in the absence or presence of LSL, the amide I band of *β*-LG was significantly enhanced, while this band of LF was decreased. This means the *β*-sheet structure of *β*-LG increased and the *β*-sheet structure of LF decreased with the addition of LSL, which is consistent with the results obtained by the CD method. As shown the arrows in Figure 3e, the peaks located at 3200~3700 cm^−1^ were attributed to the stretching vibration of hydrogen bonds [46]. This peak for *β*-LG showed a significant increase due to the formed *β*-LG/LSL complexes, indicating the formation of intermolecular or intramolecular hydrogen bonds between *β*-LG and LSL, which agrees with the results obtained from the molecular docking and fluorescence results [47].

### 3.2. Stability and Preservative Studies of LSL-Protein System

#### 3.2.1. DLS Method

Pure milk is categorized as an o/w emulsion, which is unstable. The DLS method was used to detect the particle size change with the storage time and additives. Images of milk with LSL, LF, and *β*-LG placed in a 37 °C environment after 3 days are shown in Figure 4a, and Figure 4b,c shows the mean hydrodynamic diameter (*D*_h_) at different LSL concentrations and on different days. 

For pure milk, particle size increases significantly over storage time in a 37 °C environment. On the third day, the particle size of the upper layer solution decreased due to stratification, as shown in Figure 4a(i). Generally, pure milk is unstable at this temperature. With the addition of LF and *β*-LG, the stability of the milk was significantly enhanced, and the particle size was the smallest on the third day. It is likely that the added protein was adsorbed onto the emulsion droplet, similarly to the protective effect of macromolecular compounds on the sol. For example, the combination of hyaluronic acid (HA) and kappa-carrageenan (KC) blends showed a synergistic impact on the enhanced emulsifying activity and stability of skimmed milk [48]. It is likely that one end of the macromolecular compounds was adsorbed onto the surface and completely surrounded the dispersed phase particles, as shown in the inset of Figure 4b, which played a protective role in the emulsion. In the milk system where the above proteins existed, the change of milk particle size caused by the continuous addition of LSL was investigated. It was found that the concentration of LSL had little effect on the milk particle size. However, according to the photos after each solvent had been in place for three days, the milk emulsion with LSL had no obvious particles and was dispersed homogeneously, as shown in Figure 4a(iv), which indicates that LSL has a good dispersion effect on milk emulsion.

#### 3.2.2. TTC Colony Color Test

Bacteria can reduce tetrazolium salt containing an azo group to red formazan, to form red colonies [49,50], and 2,3,5-Triphenyltetrazolium chloride (TTC) is a tetrazolium salt. Therefore, colonies will appear red after culture if bacteria are present. In this study, 8 mL lecithin Tween 80 nutrient agar was added to each petri dish, and then 1 mL 10% milk solution was added to the Nos. 2–6 petri dishes, but not to the No. 1 petri dish. Finally, 20 μL normal saline was added to the No. 2 petri dish; 10 μL 10 g/L *β*-LG + 10 μL normal saline were added to the No. 3 petri dish; 10 μL 10 g/L *β*-LG + 10 μL 10 g/L LSL were added to the No. 4 petri dish; 10 μL 10 g/L LF + 10 μL normal saline were added to the No. 5 petri dish; and 10 μL10 g/L LF + 10 μL 10 g/L LSL were added to the No. 6 petri dish. As can be seen from Figure 5a, colonies were formed after the addition of milk. According to the color reaction of the TTC solution, we counted the number of colonies under the microscope. As shown in Figure 5b, the number of colonies in the milk before and after adding *β*-LG did not change significantly, but there was a clumping phenomenon, which may be caused by a curdling effect between lactoferrin and milk. Surprisingly, the number of colonies significantly increased for the *β*-LG + LSL system. We speculated that the addition of LSL promoted the dispersion of *β*-LG and made the colonies grow evenly in the nutrient solution. However, the addition of LF had a certain antiseptic effect compared with pure milk. Moreover, the number of colonies was significantly reduced after adding LSL, which indicated that LF and LSL may have a certain synergistic effect on inhibiting the growth of colonies.

### 3.3. Molecular Docking

Molecular docking was used to study the binding environment of LSL in *β*-LG/LF, and further confirmed the above experimental results. The final data were the optimal docking results obtained from multiple docking runs. The complete molecular models of *β*-LG-LSL and LF-LSL are shown in Appendix A and the putative LSL binding sites with *β*-LG or LF are shown in Figure 6. The Δ*G* values of *β*-LG-LSL and LF-LSL were −41.11 kJ/mol and −42.85 kJ/mol, respectively, differing by 1.74 kJ/mol, which echoes the small Gibbs free energy difference gap between the two systems obtained using the Gibbs–Helmholz equation in fluorescence experiments. In addition, the amino acid residues of *β*-LG surrounded by LSL included Lys141, Leu140, Asp137, Met145, Ala26, Ile147, Ala25, Leu149, His146, Arg148, and Ser150; and the amino acid residues of LF surrounded by LSL included His91, Thr90, Ser252, Val250, Pro251, Tyr319, Gln249, Thr688, Tyr319, Ser322, Cys405, Gly406, Asp602, and Arg600. These amino acid residues were involved in the binding of LSL and *β*-LG/LF, respectively. For the *β*-LG and LSL system, the formed three hydrogen bonds (His146 1.88 Å, Arg148 2.77 Å, and 2.64 Å) confirmed that LSL could interact with *β*-LG to form a complex, mainly by van der Waals interactions and hydrogen bonds with many binding sites. The interaction force between *β*-LG and LSL obtained from the fluorescence experiments echoes this result. For the LF-LSL system, the distance between Tyr319 and LSL is 2.26 Å, which may be caused by the presence of LSL changing the microenvironment around Tyr in LF, and resulting in changes in protein conformation. This is similar to the results of the synchronous fluorescence experiments.

## 4. Conclusions

LF and *β*-LG are important components of whey proteins, and the interaction mechanism of biosurfactant-bland protein system is of great significance for the development and application of natural additives in the cosmetics and food industries. Consequently, the binding mechanisms of LSL and *β*-LG/LF were studied in detail using multispectral and molecular docking methods. It is proved that the quenching mechanism of LSL on both *β*-LG and LF was static quenching and that the formed non-covalent complexes were the result of from the different interacting forces. The biosurfactant monomers could reach the interior of the protein to form non-covalent complexes, which affected the secondary and tertiary structures of the proteins. The addition of LSL induced *β*-LG to decrease the *α*-helix and random coil structure, and to increase the *β*-sheet and *β*-turn structure, while the opposite was found in LF. By analyzing the thermodynamic parameters, it can be seen that this is an enthalpy-driven process through hydrogen bonds and the van der Waals force for the LSL-*β*-LG complex formation, while it is an entropy-enthalpy co-driven process through electrostatic force for the LSL-LF complex formation. For the preservative experiment aimed at milk, there is an obvious contrast between LF and *β*-LG. To summarize, LF had certain antiseptic abilities due to its decreased numbers of colonies, and the formed complexes with LSL further decreased the numbers of colonies, while the existence of *β*-LG or/and LSL shortened the shelf life of the milk. The stability of milk emulsion was improved in all cases, i.e., with *β*-LG, LF, or the mixed system with LSL. Although LF and *β*-LG are important components of whey protein, there are qualitative differences in the mechanism and functional properties with biosurfactants. This study shows that protein–biosurfactant mixtures can be used as a natural compound additive in industrial fields such as food and cosmetics to obtain safer products for a healthier future.

## Figures and Tables

**Figure 1 foods-12-01561-f001:**
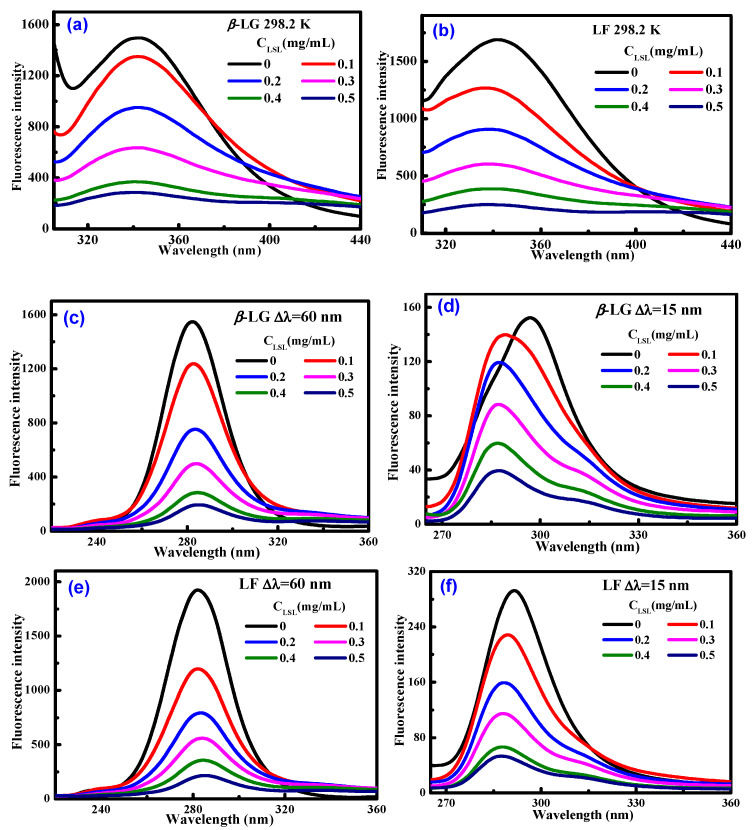
Fluorescence spectra of *β*-LG (**a**) or LF (**b**) solutions with different LSL concentrations at 298.2 K. Synchronous fluorescence spectra of *β*-LG ((**c**), Δλ = 60 nm; (**d**), Δλ = 15 nm) or LF ((**e**), Δλ = 60 nm; (**f**), Δλ = 15 nm) solutions at 298.2 K. The protein concentrations were fixed at 0.2 g/L and the LSL concentrations varied between 0~0.5 g/L.

**Figure 2 foods-12-01561-f002:**
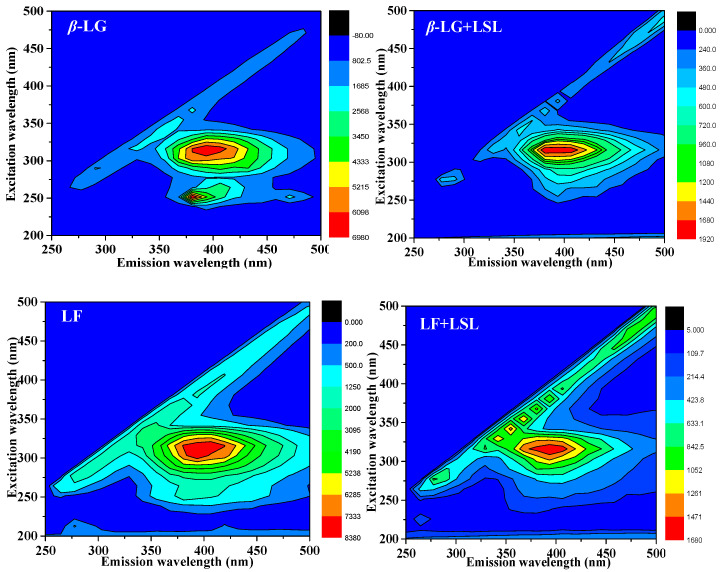
3D fluorescence spectral plots of the interaction for LSL with *β*-LG/LF. The protein concentration was fixed at 1 g/L, and the concentration ratio of protein to LSL was 5:1.

**Figure 3 foods-12-01561-f003:**
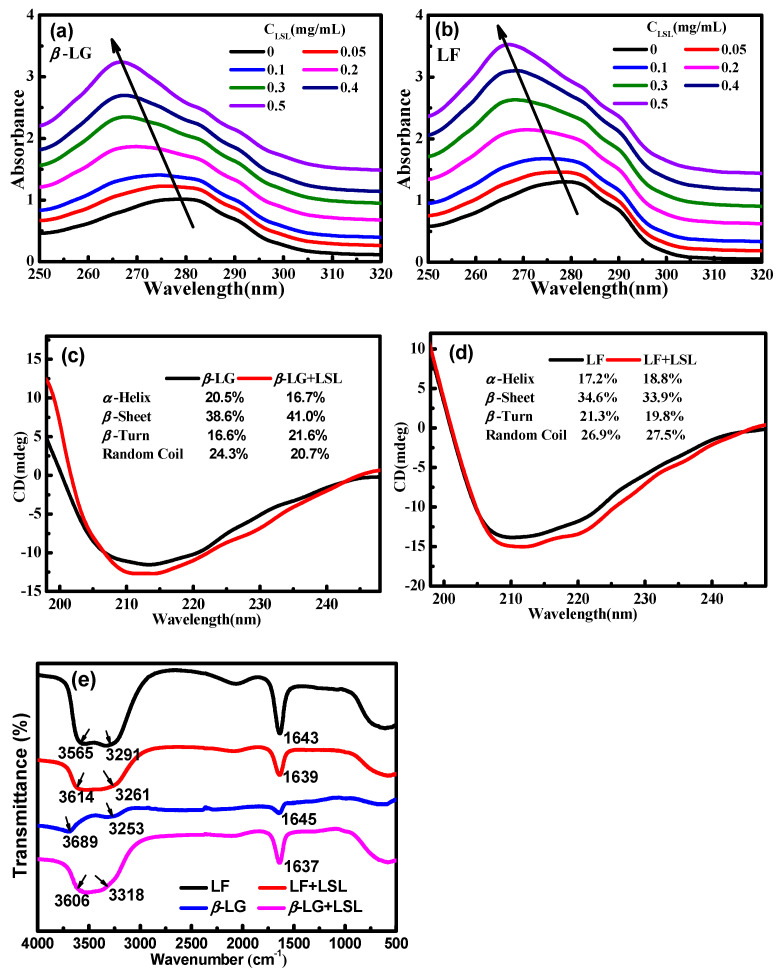
The UV-vis absorption spectra of *β*-LG (**a**) and LF (**b**) with different LSL concentrations at fixed protein concentration of 1 g/L and at 298.2 K; the CD spectra of *β*-LG (**c**) and LF (**d**) at fixed protein and LSL concentrations of 0.2 g/L and 1 g/L and at 298.2 K; the FT-IR spectra (**e**) at fixed protein and LSL concentrations of 1 g/L and 1 g/L and at 298.2 K.

**Figure 4 foods-12-01561-f004:**
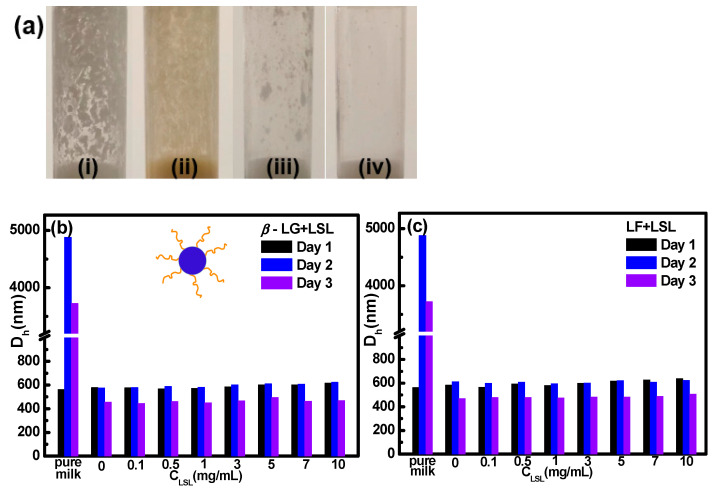
(**a**) Pictures of bottle walls of different lotions 3 days after placement. (**i**) The pure milk emulsion (**ii**) The milk emulsion with 4 g/L *β*-LG. (**iii**) The milk emulsion with 4g/L LF. (**iv**) The milk emulsion with 4 g/L LSL. (**b**) The milk particle size with *β*-LG-LSL at different LSL concentrations and at 298.2 K. (**c**) The milk particle size with LF-LSL at different LSL concentrations and at 298.2 K.

**Figure 5 foods-12-01561-f005:**
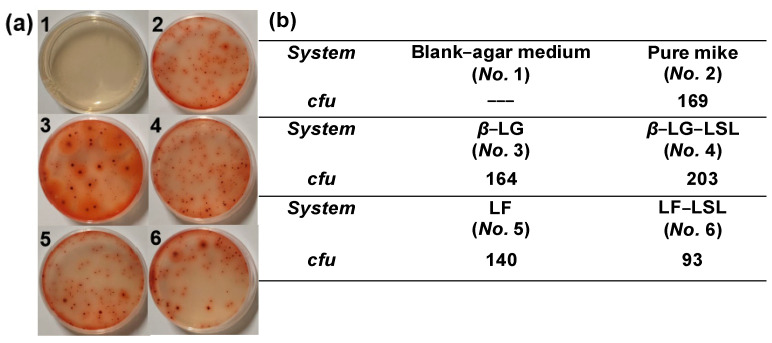
(**a**) The results of the Colony Color Test of *β*-LG/LF-LSL complex dispersion in milk emulsion at 298.2 K. (**b**) The number of colonies in *β*-LG/LF-LSL system under milk emulsion.

**Figure 6 foods-12-01561-f006:**
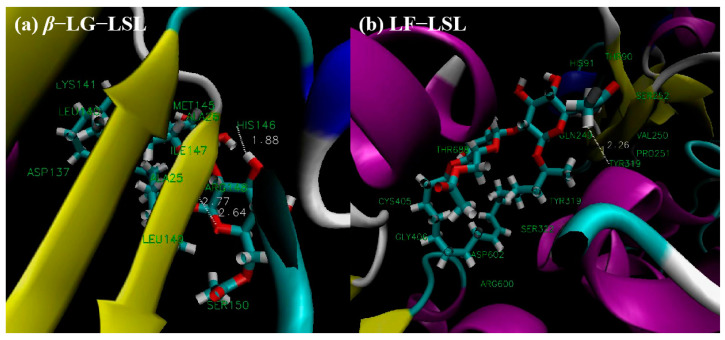
The putative LSL binding sites with *β*-LG (**a**) or LF (**b**) in 3D molecular docking analysis.

**Table 1 foods-12-01561-t001:** Fluorescence quenching constants, binding constants, and thermodynamic parameters for the interactions of *β*-LG/LF and LSL at different temperatures.

System	*T* (*K*)	*K*_SV_ (M^−1^)	*K*_q_ (M^−1^S^−1^)	*R* ^2^	*n*	*K*_a_ (M^−1^)	*R* ^2^	Δ*H*_m_ (kJ/mol)	Δ*S*_m_ [(J/(mol·k)]	Δ*G*_m_ (kJ/mol)
*β*-LG + LSL	288.2	8.183 × 10^3^	8.183 × 10^11^	0.982	2.45	31.48	0.987	−20.19	−41.36	−8.26
	298.2	7.248 × 10^3^	7.248 × 10^11^	0.983	2.31	24.00	0.992		−41.13	−7.92
	308.2	7.159 × 10^3^	7.159 × 10^11^	0.986	1.97	18.21	0.997		−41.38	−7.43
LF + LSL	288.2	8.278 × 10^3^	8.278 × 10^11^	0.984	1.82	18.22	0.997	−4.51	8.48	−6.95
	298.2	7.701 × 10^3^	7.701 × 10^11^	0.985	1.82	17.05	0.998		8.46	−7.03
	308.2	7.515 × 10^3^	7.515 × 10^11^	0.984	1.79	16.14	0.996		8.49	−7.13

**Table 2 foods-12-01561-t002:** Three-dimensional fluorescence spectral parameters of *β*-LG/LF and LSL.

System	Peak *a*		Peak *b*	
	Peak Position λ_ex_/λ_em_ (nm/nm)	Intensity *F*_0_	Peak Position λ_ex_/λ_em_ (nm/nm)	Intensity *F*_0_
β-LG	280/340	6910	230/330	6968
β-LG + LSL	280/330	1919	230/340	344.5
LF	280/340	8379	230/330	1892
LF + LSL	280/340	1676	230/330	201.5

## Data Availability

The data presented in this study are available upon request from the corresponding author.

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
