# Peer review of "Molecular Interaction Mechanism and Preservative Effect of Lactone Sophorolipid and Lactoferrin/β-Lactoglobulin Systems"

_foods, 2023, doi:10.3390/foods12081561_

Round 1

Reviewer 1 Report

Dear Food Editor, I have reviewed the manuscript entitled "Molecular Interaction Mechanism and Preservative Effect for the Systems of Lactone Sophorolipid and Lactoferrin / β-Lactoglobulin" for consideration and deemed it fit for publication after the authors make the following recommendations. My recommendation is minor revision.

1. It is important to include in the introduction part the definition and properties of a surfactant.

2. Discuss and compare each result with other similar investigations to highlight the manuscript more.

- A comprehensive research on Lactone Sophorolipid (LSL) and Soy Protein Isolate (SPI) interacting mixture. Journal of Molecular Liquids, 339, 117239.

- Kurtzman, C. P., Price, N. P., Ray, K. J., & Kuo, T. M. (2010). Production of sophorolipid biosurfactants by multiple species of the Starmerella (Candida) bombicola yeast clade. FEMS microbiology letters, 311(2), 140-146.

- Pal, S., Chatterjee, N., Das, A.K., McClements, D.J., & Dhar, P. (2023). Sophorolipids: A comprehensive review on properties and applications. Advances in Colloid and Interface Science, 102856.

3. I recommend performing FTIR analysis to compare the results with Molecular Docking

Author Response

Reply: We thank the reviewer for evaluating our research and we are really grateful for the reviewer’s affirmation of our manuscript. We have revised the manuscript in accordance with the suggestion of the Reviewer. The detailed changes are listed below.

Reviewer 2 Report

The paper is well constructed. The methods are proper for the subject. Presentation of the problem is interesting and the discussion is good. English language level is satisfactory. They observed static quenching with formation of non-covalent complexes by hydrogen bonds and van der Waals force for LSL-β-LG system and electrostatic force for LSL-LF system. After adding β-LG, LF, or their mixed system with LSL to the milk, the stability of milk emulsion was effectively improved, while the preservative ability was effective enhanced for LF or LSL-LF addition.

Several important publications are missing and they should be added to the text.

Aguirre-Ramírez et al. (2021) investigated surfactants: physicochemical interactions with biological macromolecules (Biotechnol Lett., 43(3):523-535. doi: 10.1007/s10529-020-03054-1).

An important number of macromolecules are present in mixtures with surfactants, where a combination of hydrophobic and electrostatic interactions is responsible for the specific properties of any solution. It has been demonstrated that surfactants can help the formation of helices in some proteins thereby promoting protein structure formation. On the other hand, there is extensive research towards the use of surfactants to solubilize drugs and pharmaceuticals; therefore, it is evident that the interaction between surfactants with macromolecules is important for many applications which includes environmental processes and the pharmaceutical industry. In this review, the Authors describe the properties of different types of surfactants that are relevant for their physicochemical interactions with biological macromolecules, from macromolecules–surfactant complexes to hydrophobic and electrostatic interactions.

Hansted et al. (2011) looked at the effect of protein-surfactant interactions on aggregation of β-lactoglobulin. Biochim Biophys Acta., 1814(5):713-23. doi: 10.1016/j.bbapap.2011.03.011. 

They used spectroscopic and calorimetric techniques to elucidate how anionic, cationic and non-ionic surfactants interact with bovine βLG and modulate its heat-induced aggregation. Alkyl trimethyl ammonium chlorides (xTAC) strongly promote aggregation, while sodium alkyl sulfates (SxS) and alkyl maltopyranosides (xM) reduce aggregation. Sodium dodecyl sulfate (SDS) binds to non-aggregated βLG in several steps, but reduction of aggregation was associated with the first binding step, which occurs far below the critical micelle concentration. In contrast, micellar concentrations of xMs are required to reduce aggregation. The ranking order for reduction of aggregation (normalized to their tendency to self-associate) was C10-C12>C8>C14 for SxS and C8>C10>C12>C14>C16 for xM. xTAC promote aggregation in the same ranking order as xM reduce it. They conclude that SxS reduce aggregation by stabilizing the protein's ligand-bound state (the melting temperature t(m) increases by up to 10°C) and altering its charge potential. xM monomers also stabilize the protein's ligand-bound state (increasing t(m) up to 6°C) but in the absence of charged head groups this is not sufficient by itself to prevent aggregation. Although micelles of both anionic and non-ionic surfactants destabilize βLG, they also solubilize unfolded protein monomers, leaving them unavailable for protein-protein association and thus inhibiting aggregation. Cationic surfactants promote aggregation by a combination of destabilization and charge neutralization. The food compatible surfactant sodium dodecanoate also inhibited aggregation well below the cmc, suggesting that surfactants may be a practical way to modulate whey protein properties.

Pedersen et al. (2019) presented a complete picture of protein unfolding and refolding in surfactants. Chem Sci., 22;11(3):699-712. doi: 10.1039/c9sc04831f. 

They used combined stopped-flow time-resolved SAXS, fluorescence, and circular dichroism, respectively, to provide an unprecedented in-depth picture of the different steps involved in both protein unfolding and refolding in the presence of SDS and C12E8. During unfolding, core-shell BLG-SDS complexes were formed within 10 ms. This involved an initial rapid process where protein and SDS formed aggregates, followed by two slower processes, where the complexes first disaggregated into single protein structures situated asymmetrically on the SDS micelles, followed by isotropic redistribution of the protein. Refolding kinetics (>100 s) were slower than unfolding (<30 s), and involved rearrangements within the mixing deadtime (5 ms) and transient accumulation of unfolded monomeric protein, differing in structure from the original bLG-SDS structure. Refolding of bLG involved two steps: extraction of most of the SDS from the complexes followed by protein refolding. These results reveal that surfactant-mediated unfolding and refolding of proteins are complex processes with rearrangements occurring on time scales from sub-milliseconds to minutes.

Andersen et al. (2016) investigated weak and Saturable Protein-Surfactant Interactions in the Denaturation of Apo-α-Lactalbumin by Acidic and Lactonic Sophorolipid. Front Microbiol., 8(7):1711. doi: 10.3389/fmicb.2016.01711.

They presented a study of the interactions between the model protein apo-α-lactalbumin (apo-aLA) and the biosurfactant sophorolipid (SL) produced by the yeast Starmerella bombicola. SL occurs both as an acidic and a lactonic form; the lactonic form (lactSL) is sparingly soluble and has a lower critical micelle concentration (cmc) than the acidic form [non-acetylated acidic sophorolipid (acidSL)]. They showed that acidSL affects apo-aLA in a similar way to the related glycolipid biosurfactant rhamnolipid (RL), with the important difference that RL is also active below the cmc in contrast to acidSL. Using isothermal titration calorimetry data, they observed that acidSL has weak and saturable interactions with apo-aLA at low concentrations; due to the relatively low cmc of acidSL (which means that the monomer concentration is limited to ca. 0–1 mM SL), it is only possible to observe interactions with monomeric acidSL at high apo-aLA concentrations. However, the denaturation kinetics of apo-aLA in the presence of acidSL are consistent with a collaboration between monomeric and micellar surfactant species, similar to RL and non-ionic or zwitterionic surfactants. Inclusion of diacetylated lactonic sophorolipid (lactSL) as mixed micelles with acidSL lowers the cmc and this effectively reduces the rate of unfolding, emphasizing that SL like other biosurfactants is a gentle anionic surfactant.

Author Response

Reply: We thank the reviewer for evaluating our research and we are really grateful for the reviewer’s affirmation of our manuscript. We are greatly touched by the reviewer’s comments, which made a detailed summary of the necessary reference. We have downloaded these important publications, and cited them after carefully reading.
